# Methods of Pre-Clustering and Generating Time Series Images for Detecting Anomalies in Electric Power Usage Data

Sangwon Oh [1], Seungmin Oh [1], Tai-Won Um [2], Jinsul Kim [1,*] and Young-Ae Jung [3,*]

1  Department of ICT Convergence System Engineering, Chonnam National University, Gwangju 61186, Korea
2  Graduate School of Data Science, Chonnam National University, Gwangju 61186, Korea
3  Division of Information Technology Education, Sunmoon University, Asan 31460, Korea
*  Correspondence: jsworld@jnu.ac.kr (J.K.); dr.youngae.jung@gmail.com (Y.-A.J.); Tel.: +82-62-530-1808 (J.K.); +82-41-530-2420 (Y.-A.J.)

**Abstract:** As electricity supply expands, it is essential for providers to predict and analyze consumer electricity patterns to plan effective electricity supply policies. In general, electricity consumption data take the form of time series data, and to analyze the data, it is first necessary to check if there is no data contamination. For this, the process of verifying that there are no abnormalities in the data is essential. Especially for power data, anomalies are often recorded over multiple time units rather than a single point. In addition, due to various external factors, each set of power consumption data does not have consistent data features, so the importance of pre-clustering is highlighted. In this paper, we propose a method using a CNN model using pre-clustering-based time series images to detect anomalies in time series power usage data. For pre-clustering, the performances were compared using k-means, k-shapes clustering, and SOM algorithms. After pre-clustering, a method using the ARIMA model, a statistical technique for anomaly detection, and a CNN-based model by converting time series data into images compared the methods used. As a result, the pre-clustered data produced higher accuracy anomaly detection results than the non-clustered data, and the CNN-based binary classification model using time series images had higher accuracy than the ARIMA model.

**Keywords:** electric power data; clustering; CNN; time series image; anomaly detection





## 1. Introduction

Most people living in modern civilization are provided with electricity produced by the state or private enterprise. From the perspective of the private company or government that supplies electricity, it is important to manage electric power received so that it can be used efficiently. From the perspective of providing electricity, it is necessary to understand how customers use the electric power they receive to make plans to reduce various costs in terms of production and supply [1,2]. Therefore, analyzing data on how customers use electric power has become an important task for most electric power providers [3].

In general, data representing the electric power used by customers are stored and recorded in a time series [4,5]. This is because the detector recodes electric power usage according to the time unit. These data obtained through this process have various features, such as cycles and trends over time. However, detection devices for calculating the electric power used by customers can lose data uniformity due to various environmental factors. Considering possible electric power system component failures, communication failures, protection failures, and market and load uncertainties, it is important to analyze recorded power data by taking errors resulting from various factors into account [6–8].

Detection of anomalies in power usage data is a task that can monitor whether power is wasted. Most power usage data take the form of time series data. The usage is recorded according to the unit of time, and they are gathered to show one periodic characteristic. In these cases, anomalies are measured using a variety of statistical techniques [9,10] or simple LSTM-AE family models [11–13]. However, if it is not a point anomaly in which anomalies

are recorded in only a single time unit, but a collective anomaly in which anomalies are accumulated over a day or more, the effect of statistical techniques is not greater than before. Since power usage data have periodicity, context should also be considered when it comes to anomalies. So, most anomalies in power usage data have contextual properties. The contextual anomaly has many variables to refer to and it is difficult to know which variables to refer to. Therefore, it is worthwhile to try different conditions and methodologies to detect anomalies in power usage data [14–16].

In addition, since electric power detection devices are provided to individuals, data belonging to a cluster corresponding to a unit (building, apartment complex, region) are not subjected to the same environmental change, making it difficult to cluster. If common environmental variables are applied in clusters, data features can be easily extracted by using a single method of clustering. However, since data with various errors are combined and delivered, it is important to cluster data to which individual environmental variables are applied. Studies have been conducted using various clustering methods or using clustering with complex process forms [17], such as clustering data from electrical power distribution systems using a feature-based clustering approach that performs principal component analysis first [18]. By using a clustering method to detect anomalies in power usage data, capturing features of recorded data according to environmental variables can be advanced.

To detect anomalies in the power usage time series data recorded according to various environmental variables, a pre-clustering method can be used in the data preprocessing step, and various methods other than the anomaly detection method using the existing LSTM and statistical techniques can be used. In this paper, we used a CNN-based binary classification model that is different from existing methods to preprocess time series data with various pre-clustering methods and to detect effective anomalies. To apply the CNN-based model to time series data, methods for generating time series images were applied. If the approach proposed in this paper is used, we can show high performance in electric power analysis tasks such as anomaly detection of power data in various electric power detection systems. For example, in a smart grid, a pre-clustering-based power anomaly detection method can be used to have a generation-optimized electric power supply pattern. Moreover, the method of this study can be adapted to label the generation in which anomalies are recorded.

The rest of this paper is organized as follows. Section 2 introduces time series data, an important keyword appearing in this paper, the clustering method used in this experiment, and methods for generating time series images. Section 3 shows the overall clustering process and anomaly detection method used in the experiment. Experimental results and conclusions are summarized in Sections 4 and 5, respectively.

## 2. Related Research

This section introduces the algorithms and concepts mentioned and used in this study. How electric power data are collected and pre-processed, general-purpose clustering algorithms for time series data, and generating time series image algorithms used in this study are described in this section.

### 2.1. Electric Power Data

Large-scale electric power suppliers provide power systems to private customers, supply current separately, and provide services in the form of receiving cost for the usage. Providers generate electric power through power plants, build facilities to supply power to customers, sell power to customers, and record calculated data. Electric power data are classified into three categories according to the place and time when data are recorded. The first is power generation statistics, which record the amount of power produced by power plants. The second is facility statistics, which show the trends and status of facilities built to supply power to private users. The third is sales statistics, which supply power to users and record how much power they lose during supply.

Power data are preprocessed from raw data into various types of data. Image data are used for visually expressing power facilities, faults, or defects. For example, we can use a picture of a detector or a faulty power plant equipment as image data. Financial data are used for expressing financial characteristics such as unit price and monthly sales volume to establish a supply plan. Financial data include data that adjust unit price to plan power supply according to the recorded amount of electric power. Time series data are used for recording data in which private customers use power over time or receive power from power plants. Time series data can be used to perform a variety of tasks, such as making predictions and detecting anomalies [19]. In general, data such as power supply and demand, power demand, and power loss are used to predict the power consumption of private customers and to cluster power demand patterns. Since all data are recorded and collected over time, time series data are the basic type of electric power data because it is easy to preprocess such data. Power suppliers need to understand features of electric power data by clustering collected data to efficiently establish power supply and facility maintenance plans.

Figure 1 is an example of normalized power data actually detected in a building. We can clearly see the trend in electric power usage from day to day, and if we take a long time unit as a basis, we can see the trend in that electric power usage. These data are recorded by a general building detector, and if the detector malfunctions due to various environmental factors, the maximum or minimum value of the measured power value may be recorded abnormally.

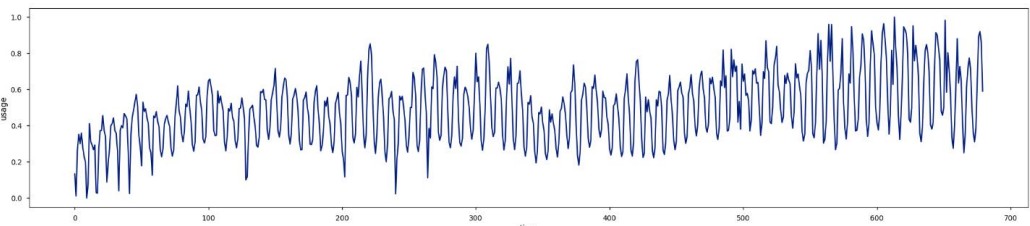

**Figure 1.** Example of normalized electric power usage data.

### 2.2. Time Series Data Clustering

The goal of clustering is to identify structure in an unlabeled dataset by objectively organizing data into homogeneous groups where the within-group-object similarity is minimized, and the between-group-object dissimilarity is maximized. Unlike static data, which have certain rules and fixed format, time series data have a feature in that values will change with time. Given a set of unlabeled time series, it is often desirable to determine groups with similar time series.

Figure 2 shows three different approaches: raw-data-based, feature-based, and model-based. Note that the left branch of the model-based approach trains the model and uses model parameters for clustering without needing another clustering algorithm. Unlike a raw-data-based approach, a feature-based approach has steps to extract features, allowing researchers to effectively understand characteristics of the data. A model-based approach clusters by extracting model parameters or determining coefficients and residuals through various models. This approach allows researchers to proceed with the clustering process and use several general-purpose clustering algorithms to cluster according to the given data [20–22].

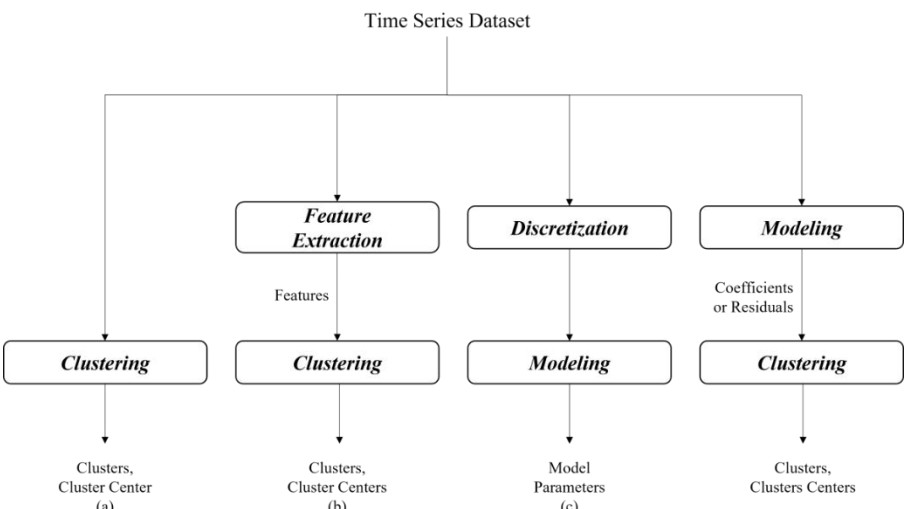

**Figure 2.** Three time series clustering approaches: (**a**) raw-data-based; (**b**) feature-based; (**c**) model-based.

### 2.2.1. Agglomerative Hierarchical Clustering

An agglomerative clustering method can repeatedly perform the following commands. At startup, each data coordinate is designated as one cluster and two most similar clusters are merged. At that time, similar clusters are combined until the specified number of clusters remains. There are three main ways to merge two clusters. The first is Ward, which merges two clusters so that the increase of variance within all clusters is the smallest. The second is average, which combines two clusters with the shortest average distance between cluster centers. The third is complete, which combines two clusters with the shortest maximum distance between cluster points.

Cluster aggregations each stage or level are grouped into cluster trees and operated, forming a layer of trees. As shown in Figure 3 below, agglomerative or divisive strategies are performed depending upon whether a bottom-up or top-down strategy is used. The agglomerative hierarchical clustering method is more popular than the divisive method. Hierarchical clustering is not restricted to cluster time series with equal length. It is applicable to series of unequal length as well if an appropriate distance measure such as DTW (dynamic time warping) is used to compute similarity.

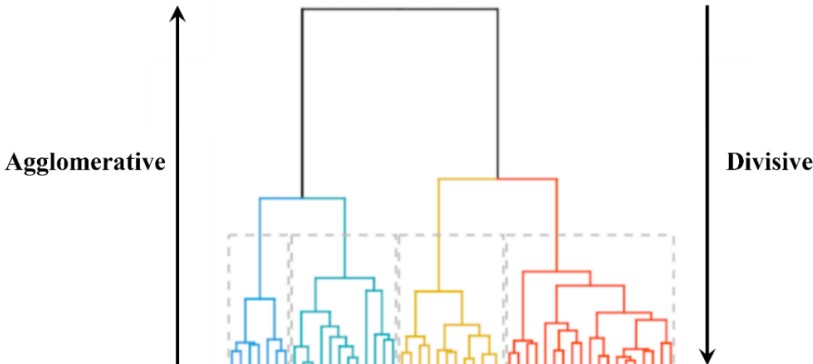

**Figure 3.** Agglomerative and divisional strategies of hierarchical clustering.

### 2.2.2. K-Means and Fuzzy c-Means

K-means (interchangeably called c-means in this study) was first developed more than three decades ago. The main idea behind it is the minimization of an objective function, which is normally chosen to be the total distance between all patterns from their respective cluster centers [23–25]. The centroid is randomly determined by the initial value of k. After

that, the process of touring all data is repeated to assign each dataset to the cluster to which the nearest centroid belongs. The centroid then moves to the center of the cluster. Algorithms alternate until the value of the objective function can no longer be reduced.

Given $n$ patterns $\{x_k | k = 1, \ldots, n\}$, c-means determine $c$ cluster centers $\{v_i | i = 1, \ldots, c\}$, by minimizing the objective function given as:

$$
\begin{aligned}
\min J_1(U, V) &= \sum_{i=1}^{c} \sum_{k=1}^{n} u_{ik} \|x_k - v_i\|^2 \\
u_{ik} &\in \{0, 1\} \forall i, k. \\
\sum_{i=1,c} u_{ik} &= 1, \ \forall k
\end{aligned}
\tag{1}
$$

where $\| \cdot \|$ in the above equation is normally the Euclidean distance measure. However, other distance measurements can also be used. Recurring solution procedures typically have the following steps:

1.   Choose $c$ $(2 \leq c \leq n)$ and $\varepsilon$ (number for stopping the iterative procedure). Set the counter $l = 0$ and the initial cluster center, $V^{(o)}$, arbitrarily.
2.   Distribute $x_k$, $\forall k$ to determine $U^{(l)}$ such that $J_1$ is minimized. This is achieved normally by reassigning $x_k$ to a new cluster that is closest to it.
3.   Modify cluster centers $V^{(l)}$.
4.   Stop if the change in $V$ is smaller than $\varepsilon$; otherwise, increment $l$ and repeat steps 2–3.

Dunn has devised a method using fuzzy segmentation by expanding the k-means algorithm and presenting the objective function as shown in Equation (2) below [26,27]:

$$
\begin{aligned}
\min J_2(U, V) &= \sum_{i=1}^{c} \sum_{k=1}^{n} (\mu_{ik})^2 \|x_k - v_i\|^2 \\
u_{ik} &\in \{0, 1\} \forall i, k, \ \sum_{i=1,c} u_{ik} = 1, \ \forall k. \\
0 &< \sum_{k=1,n} \mu_{ik} < n, \ \forall i.
\end{aligned}
\tag{2}
$$

Note that $U = [\mu_{ik}]$ in this equation and the following equations denotes the matrix of a fuzzy c-partition (same as k-partition). Fuzzy c-partition constraints are conditions under the objective function of Equation (2). In other words, each $x_k$ could belong to more than one cluster with each belongingness taking a fractional value between 0 and 1. Bezdek has generalized $J_2(U, V)$ to an infinite number of objective functions. The new objective function subject to the same fuzzy c-partition constraints is shown in Equation (3) below:

$$
\min J_m(U, V) = \sum_{i=1}^{c} \sum_{k=1}^{n} (\mu_{ik})^m \|x_k - v_i\|^2
\tag{3}
$$

By specifying the number of clusters and weighting coefficient, alternative optimization procedures are repeated to solve the fuzzy c-means model. The fuzzy c-means algorithm works better for time series with the same length. This is because the concept of the cluster center is unclear when the same cluster contains different time series of different lengths [28–30]. Various methods using fuzzy and k-means algorithms have been applied and developed for clustering [31–33].

### 2.2.3. Self-Organizing Maps (SOM)

Self-organizing maps developed by Kohonen [34] are a type of neural network in which neurons are arranged in a low-dimensional structure and trained by repetitive unsupervised learning or self-organizing procedures. The weight vector of the neuron is initialized. After presenting the pattern, it is updated according to the distance between the input pattern and the weight vector. This is in accordance with the following rule:

$$
w_i(l + 1) = \begin{cases} w_i(l) + \alpha(l)[x(l) - w_i(l)], & \text{if } i \in N_t(l) \\ w_i(l), & \text{if } i \notin N_t(l) \end{cases}
\tag{4}
$$

As neighboring neurons are updated at each stage, neighboring neurons in the network tend to indicate neighboring locations in feature spaces. Like the k-means and fuzzy c-means algorithms, SOM is an inappropriate algorithm for time series whose lengths are not the same because it defines dimensions of the weight vector [35].

In addition to methods describe above, there are various methods for clustering time series data. There is an approach that has an algorithm suitable for applying features of a time series rather than static data. However, if data are recorded with complex environmental variables such as electric power data, it is difficult to define similar time series groups and extract their characteristics. Thus, it is often desirable to learn the characteristics of time series groups using multiple clustering techniques or adopting clustering using neural network techniques. Among various neural network techniques, a method of estimating subspaces and generating sample clustering using adversarial learning effectively analyzed features and clustered data of multidimensional and multivariate data.

### 2.2.4. K-Shapes

The k-means algorithm classified clusters according to the homogeneity between individual observations using Euclidean distance (ED) to measure the distance between the data and the center of the cluster. Unlike that, the k-shape algorithm proposed by John Paparrizos [36] uses a distance measurement method called CC (cross-correlation). When two data are given in the form of a sequence, if an $l$ of padding is given in one direction, much leading occurs. Considering that the lengths of the two sequences are the same as $m$ and the maximum case of possible readings, the lengths of the sequences are $2m - 1$. Currently, the purpose of CC is to find a shifting point that gives padding by $2m - 1$ and maximizes similarity when calculating ED. According to the convolution theory, CC can be defined as Equation (5).

$$CC_w\left(\vec{x}, \vec{y}\right) = F^{-1}F\left(\vec{x}\right) * F\left(\vec{y}\right) \tag{5}$$

Since fast forwarder transform (FFT) can be used to reduce the time complexity of obtaining CC, CC is defined using the above equation. Additionally, to increase performance, the k-shape algorithm normalizes CC. This is called normalized cross-correlation (NCC). The k-shape algorithm uses the objective function shown in Equation (6) below.

$$1 - \max_w\left(\frac{CC_w\left(\vec{x}, \vec{y}\right)}{\sqrt{R_0\left(\vec{x}, \vec{x}\right) \times R_0\left(\vec{y}, \vec{y}\right)}}\right) \tag{6}$$

Since it means that the two-sequence data are close as the NCC (normalized cross-correlation) increases, the value obtained by subtracting the maximum NCC from 1 is used as the objective function. The k-shapes algorithm used CC, a distance measurement method that can be calculated from sequence data, not ED used by k-means. K-shapes show excellent performance when using sequence-type data, i.e., time series data, because they used approaches such as the dynamic time warping (DTW) method of finding points that correspond to the mean value.

### 2.3. Generating Time Series Images
### 2.3.1. Recurrence Plot (RP)

The recurrence plot algorithm is a visualization algorithm that aims to explore the m-dimensional phase space trajectory by expressing the regression of data values in two dimensions [37,38]. To convert time series data into images using RP, the m-dimensional spatial trajectory of time series data must first be configured. Given the time series data as $X = \{x_1, x_2, \ldots, x_i\}$, the m-dimensional spatial trajectory can be defined as Equation (7) below.

$$S = \{s_1 = (x_1, x_2), s_2 = (x_2, x_3), \cdots, s_n = (x_n, x_{n+1})\} \tag{7}$$

$s_n$ can be seen as the trajectory of time series data from $x_n$ to $x_{n+1}$. To express trajectory information in the form of a matrix, a distance matrix may be defined based on spatial trajectory data. To express this mathematically, the elements of the matrix can be defined as Equation (8) below.

$$R_{i,j} = \vartheta\left(\varepsilon - \|\vec{s_i} - \vec{s_j}\|\right) \tag{8}$$

$\varepsilon$ is the distance threshold, and $\theta(x)$ is the unit step function. $R_{i,j}$ is the element value of the distance matrix and means the distance ($L_2$ Norm) between $S_i$ and $S_j$ in the equation. Thus, the distance matrix $R$ is represented by a matrix of distances between the two $s$. The diagonal elements of $R$ become zero because they define the distance from themselves. Additionally, $R_{i,j}$ and $R_{j,i}$ are the same according to the definition of Equation (2) and thus become symmetric matrices. The time series image converted by the RP algorithm describes a collection of time pairs at the same location of orbit. If that orbit is strictly periodic through period $T$, then all pairs of times are separated by multiples of $T$ and displayed diagonally. That is, the visual representation of the RP provides information on the period and the change width of the time series data.

### 2.3.2. Gramian Angular Field Algorithm

A Gramian Angular Field (GAF) is an algorithm that expresses temporal correlation between each time point based on polar coordinates. Polar coordinate-based matrices have the advantage of preserving time correlations when changing time series data into images [39]. Since the GAF may first have a given time series data value too large or too small, it is normalized to the interval $[-1, 1]$ or $[0, 1]$. The time index $t_i$ of the normalized signal $\widetilde{x}_i$ is expressed in radius $r_i$, and is converted into polar coordinates as shown in Equation (9) below with the angle $\phi$.

$$\begin{cases} \phi_i = \cos^{-1}(\widetilde{x}_i), & -1 \leq \widetilde{x}_i \leq 1, \quad and \ \widetilde{x}_i \in \widetilde{X} \\ \qquad r_i = \frac{t_i}{N}, \ t_i \in N \end{cases} \tag{9}$$

Here, $N$ is the normalization constant for the range of polar coordinate systems. GF is divided into two ways depending on the sum and difference of angles. The Gramian angular summary field ($GASF$) is expressed as the sum of the angles of polar coordinate time series data consisting of time pairs of $i$ and $j$, and then is defined as Equation (10).

$$GASF = \left[\cos(\phi_i + \phi_j)\right] = \widetilde{x}' \cdot \widetilde{x} - \sqrt{I - \widetilde{x}'^2} \cdot \sqrt{I - \widetilde{x}^2} \tag{10}$$

Gramian angular difference field ($GADF$) is defined as Equation 5 of the difference between the angles of the polar coordinate system as opposed to $GASF$.

$$GADF = \left[\sin(\phi_i - \phi_j)\right] = \sqrt{I - \widetilde{x}'^2} \cdot \widetilde{x} - \widetilde{x}' \cdot \sqrt{I - \widetilde{x}^2} \tag{11}$$

GAF preserves time dependence because time increases as it moves from the top left to the bottom right. Since the main diagonal line includes the value and angle information of the raw data, the raw data may be recovered using the same.

### 2.3.3. Markov Transition Field Algorithm

The Markov transition field (MTF) algorithm is an algorithm representing the probability of transition of time series data that is discretized [39]. To configure the MTF, the given time series dataset $X$ is divided into $Q$ sections according to the value, and then assigned to the interval $q_j(j \in [1, Q])$ corresponding to the time series data value $x_i$ of the time index $t_i$. In the first Markov chain method along the time axis, a weighted adjacency matrix W of the size of $Q \times Q$ is constructed as shown in Equation (12) below.

$$W = \begin{pmatrix} w_{11|\ P(x_t \in q_1 | x_{t-1} \in q_1)} & \cdots & w_{1Q|\ P(x_t \in q_1 | x_{t-1} \in q_Q)} \\ \vdots & \ddots & \vdots \\ w_{Q1|\ P(x_t \in q_Q | x_{t-1} \in q_1)} & \cdots & w_{QQ|\ P(x_t \in q_Q | x_{t-1} \in q_Q)} \end{pmatrix} \tag{12}$$

$w_{i,j}$ represents the frequency of transition from the $q_i$ interval to the $q_j$ interval. By normalizing the sum of each column of $W$ to 1, the Markov transition matrix can be constructed. In this process, $W$ eliminates the distribution of $X$ and the time dependence on the time index $t_i$. To overcome the information loss of $W$, MTF is defined as Equation (13) below by aligning each probability along the time order.

$$M = \begin{bmatrix} M_{11} & \cdots & M_{1n} \\ \vdots & \ddots & \vdots \\ M_{n1} & \cdots & M_{nn} \end{bmatrix} = \begin{bmatrix} w_{ij|x_1 \in q_i, x_1 \in q_j} & \cdots & w_{ij|x_1 \in q_i, x_n \in q_j} \\ \vdots & \ddots & \vdots \\ w_{ij|x_n \in q_i, x_1 \in q_j} & \cdots & w_{ij|x_n \in q_i, x_n \in q_j} \end{bmatrix} \tag{13}$$

$M_{ij||i-j|=k}$, which is the i-row j-column value of MTF, represents the probability of transitioning from the interval $q_i$ to which the data value of time index $t_i$ belongs, to the interval $q_j$ to which the data value of time index $t_j$ belongs. That is, the probability of transition between points where the difference is between the two times k. If the width of the section is large, most of the values are counted as the section closest to the average, and if the width of the section is small, the value aggregated in the extreme section is reduced [39].

## 3. Experiment

This section describes the pre-clustering and anomaly detection methods proposed in this study. After explaining the data used in the experiment in detail, the experiment process according to the clustering algorithm and anomaly detection method used in the experiment is described.

### 3.1. Dataset

In this study, time series power data are clustered to predict and analyze power data. The dataset used in this study is data that store power consumption provided by the Korea Energy Agency. The electric power usage of 60 buildings was recorded every hour from 1 June 2020 to 24 August 2020. The column of the dataset is configured as shown in Table 1 below. There are a total of 122,400 rows in the dataset, and no outliers exist. Figure 4 shows a boxplot for each column in the dataset. Outliers were recorded in the wind speed column and humidity column in all buildings, and few outliers were found in power usage or temperature. In this study, the performance of time series data prediction and analysis according to the presence or absence of clustering was compared using the above dataset. Columns other than electric power usage were not used to measure performance under the same conditions and to consider the characteristics of power data. This experiment compared the performance according to the clustering technique performed in the preprocessing process to predict the power consumption dataset in a time series data format.

**Table 1.** Electric power dataset description.

| Column Name | Data Type | Min | Max | Mean | Standard Deviation |
|---|---|---|---|---|---|
| Building Number | Categorical Data | 1 | 60 | - | - |
| Date time | Date Time | 1 June 2020, 00 h | 24 August 2020, 23 h | - | - |
| Power Usage (kWh) | Float | 0.0000 | 17,739.2250 | 2324.8309 | 2058.9993 |
| Temperature (°C) | Float | 11.1000 | 36.3000 | 24.2517 | 3.4079 |
| Wind Speed (m/s) | Float | 0.0000 | 20.1000 | 2.1516 | 1.5148 |
| Humidity (%) | Float | 19.0000 | 100.0000 | 80.1698 | 15.5259 |
| Precipitation (mm) | Float | 0.0000 | 81.5000 | 0.5150 | 2.6245 |
| Sunlight (h) | Float | 0.0000 | 1.0000 | 0.2135 | 0.3705 |
| Operation of non-electric cooling equipment | Boolean | - | - | - | - |
| Solar Power | Boolean | - | - | - | - |

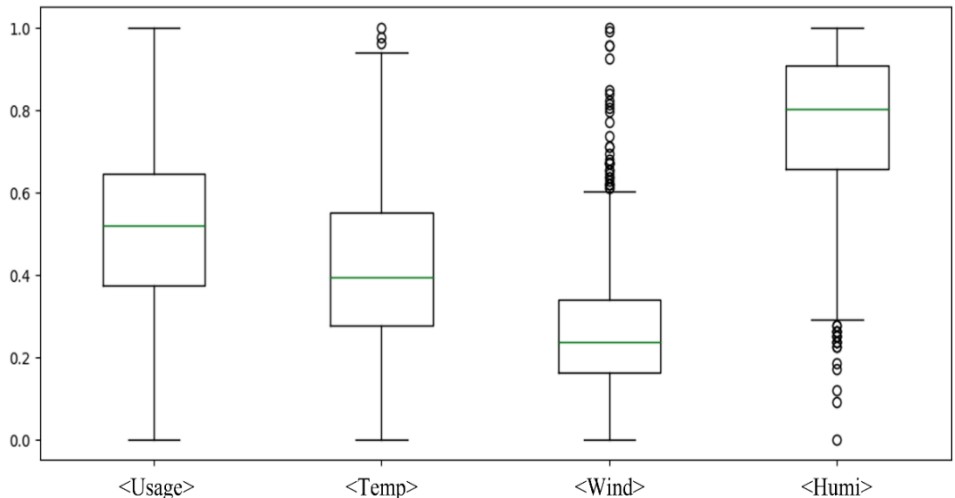

**Figure 4.** Boxplot of electric power dataset for a specific building.

### 3.2. Clustering Methods

In the pre-clustering step, 60 building power usage data were clustered. To detect anomalies, time series data were clustered to apply the time series feature of each cluster. For detecting an anomaly, we clustered using four methods to explore effective clustering methods. The first was to use only the power usage column. Anomaly detection was performed using only single-dimensional data without separate categorical data. Second and third, power usage data were clustered using the k-means and k-shapes algorithm and separate categorical data were added and used. To set k, clustering was performed from 1 to 60, the maximum number of buildings, and the appropriate k was set using the elbow method. As a result, it was appropriate to set k to 7 as shown in Figure 5. The clustering results using the k-shape algorithm can be confirmed in Figure 6. Fourth, power usage data were clustered using the SOM algorithm and separate categorical data were added and used. To determine the optimal number of nodes and map arrangement, appropriate nodes were obtained using Equation (14) proposed by Tian et al.

$$M \approx 5\sqrt{N} \tag{14}$$

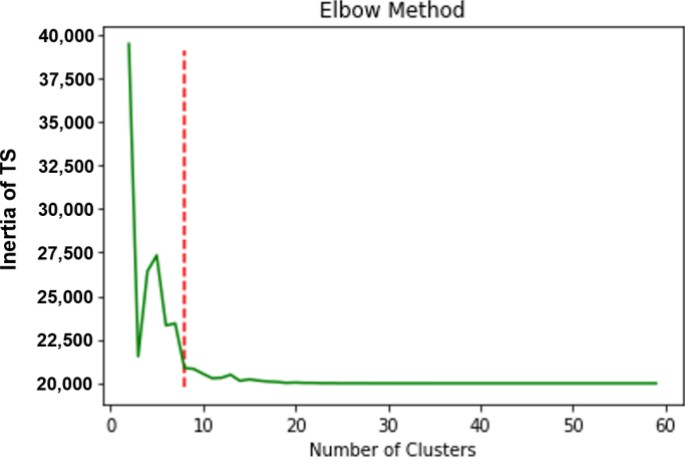

**Figure 5.** Result of elbow method to set k.

$N$ is the number of objects to be observed and $M$ is the number of neurons, which is an integer close to the value of the right-hand side. When the observation target is 60 (number of builds), the number of neurons calculated was about 39 $(= 5\sqrt{60})$. Therefore,

clustering was performed using SOM composed of a feature map of $7 \times 6$ corresponding to the number of arrays close to the number of neurons.

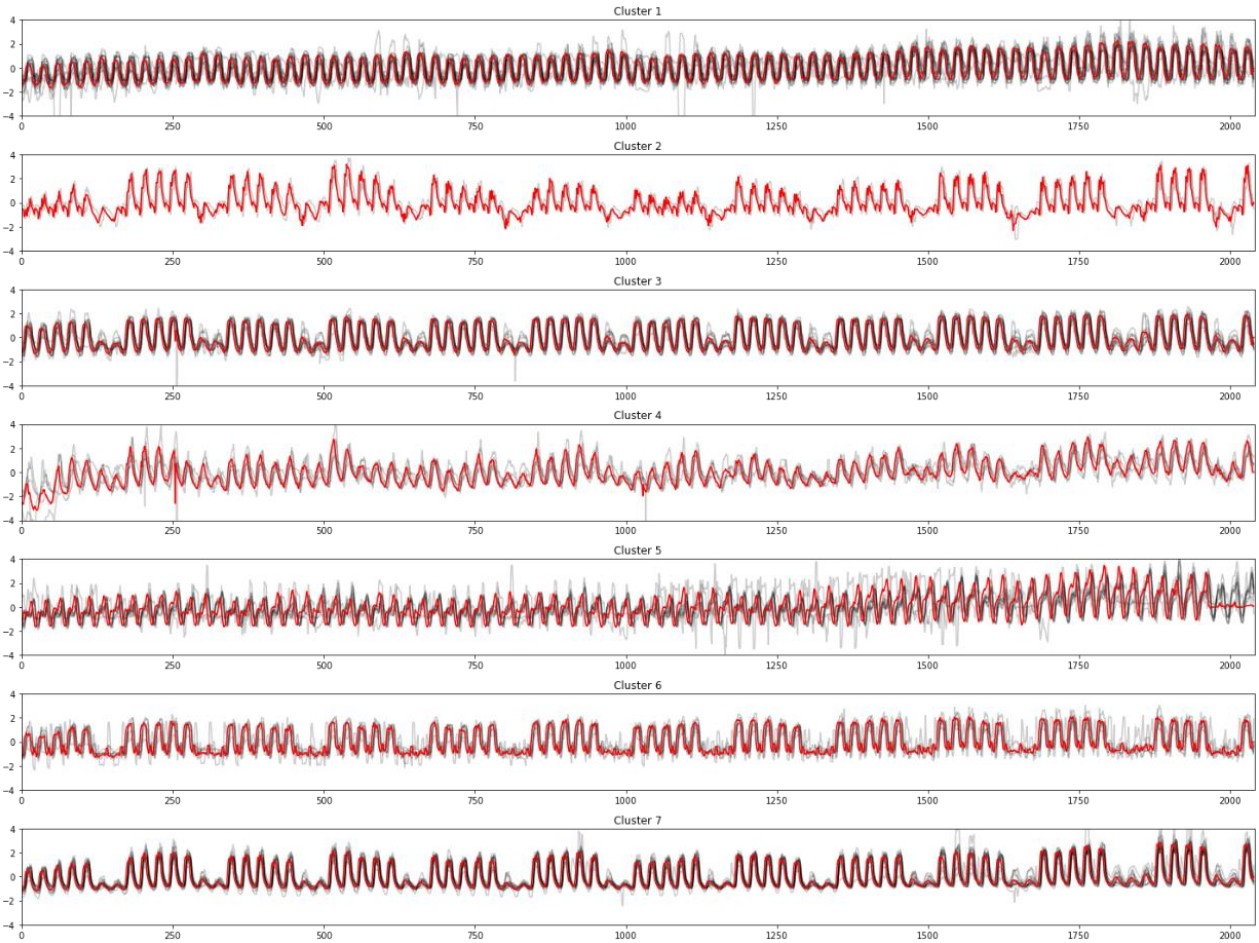

**Figure 6.** K-shape clustering result.

### 3.3. Data Preprocessing and Add Anomaly for Learning

We needed to preprocess the data to detect whether anomalies were recorded on a daily basis within a total of 60 building power usage data belonging to a certain number of clusters. For each building, the power usage data were recorded every hour for 85 days. Building power usage data were divided into 85 sections to extract daily power usage data. In conclusion, power usage data having a shape of (60, 2040) were preprocessed into data having a shape of (60, 85, 24). For example, if there were 12 building power usage data in the first cluster, these data were converted into time series data with a shape of (12, 85, 24).

Anomalies were not recorded in the dataset used in this experiment. As shown in Figure 7, we had a process of arbitrarily injecting anomalies to construct a training model that detects anomalies through supervised learning. First, the daily power usage data were normalized. Anomalies were injected by converting any data from normalized daily power usage data to values close to the maximum and minimum values in daily data. Additionally, the index of the data with injected anomalies was extracted and set as the label to be used in supervised learning. In conclusion, anomalies were injected into random indexes, and this label was set to ground-truth and used as a y dataset for the model trained in this experiment.

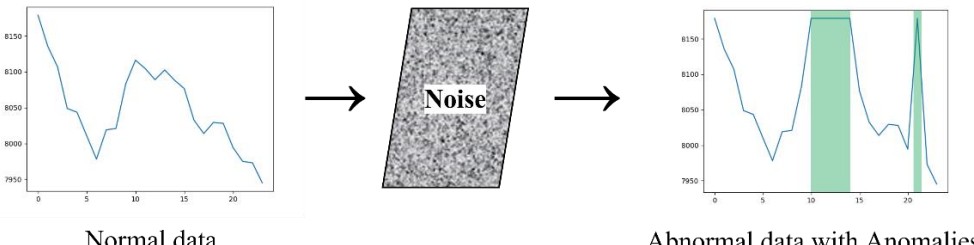

Normal data                Abnormal data with Anomalies

**Figure 7.** Process of injecting anomalies in data.

### 3.4. Using ARIMA for Detecting Anomalies

We adopted a method using ARIMA that statistically analyze time-series data, to validate and compare the performance of imaging time-series data for anomaly detection. The optimal p, d, and q factors were searched by applying the ARIMA model to the preprocessed power usage time series data. As a result of the search, we confirmed that (p, d, q) has the most appropriate ARIMA model when (4, 1, 1). In the analyzed time series data, the date on which data having a difference value of 10 or more is recorded is defined as data on which anomalies are recorded. As shown in Figure 8 below, after visualizing the data at the time when the anomalies were recorded, the date containing the data was anomaly defined and an F1-score was given compared to the label.

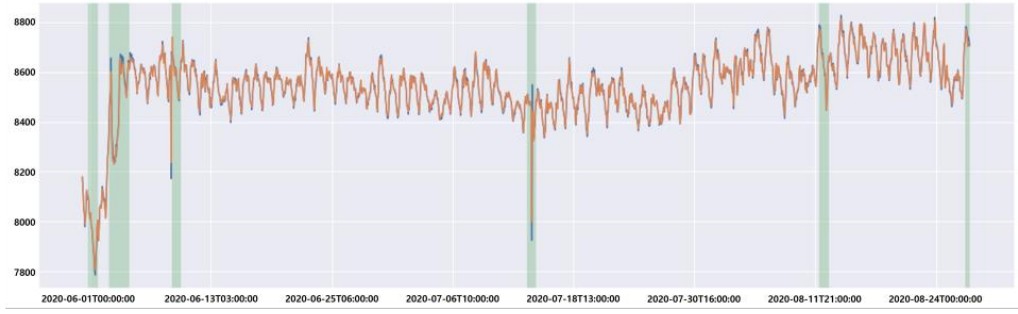

**Figure 8.** Anomaly detection result using ARIMA in building power usage data.

### 3.5. Generating Time Series Images

In this experiment, as shown in Figure 9, time series images were generated using RP, GASF, GADF, and MTF algorithms for time series data recorded in 24 h time units. When using the RP algorithm, the cross recurrence plot generates a black and white time series image by substituting 1 if it is greater than a certain distance (element value), but in this experiment, the distance threshold was not used to obtain more information from data in the CNN model. CNN-based layers were designed as shown in Table 2 with the generated time series image to learn a model that classifies normal data and anomalies.

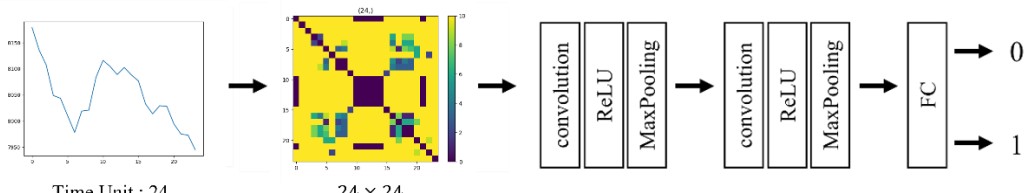

**Figure 9.** Proposed detecting anomaly method using time series images.

**Table 2.** CNN-based model structure for binary classification.

| Layer | Channel | Kernel |
|---|---|---|
| Input | $1 \times 24 \times 24$ | $1 \times 1 \times 24$ |
| Conv 1 | $24 \times 24$ | 3 |
| ReLU | $24 \times 24$ | 3 |
| MaxPooling | $24 \times 24$ | 3 |
| Conv2 | $16 \times 16$ | 3 |
| ReLU | $16 \times 16$ | 3 |
| MaxPooling | $16 \times 16$ | 3 |
| Fully Connected | 1 | N/A |

Then, as shown in Figures 10–13, time series data recorded for 85 days were imaged. Time series data recorded over 85 days per building were imaged using 85 recurrence plot (RP), GASF, GADF, and MTF algorithms. The time series image coordinate values calculated through each algorithm are expressed as Figures 10–13 through colormap. In Figures 11–13, the colormap used in this figure is a jet, which set a minimum value to blue and a maximum value to red, so the coordinate values are converted into colors in RGB format in jet colormap. Exceptionally, in Figure 10, we used a specific colormap, which set a minimum value to blue and a maximum value to yellow. Data with existing shapes (60, 85, 24) have been increased to (60, 85, 24, 24) shapes. The two results were compared using original data and dimensionally increased data as training and testing data for anomaly detection models. As ARIMA models were used to detect anomalies for each clustered time series data, CNN-based models can be used to classify whether they were anomalies by converting clustered power usage data into time series image datasets.

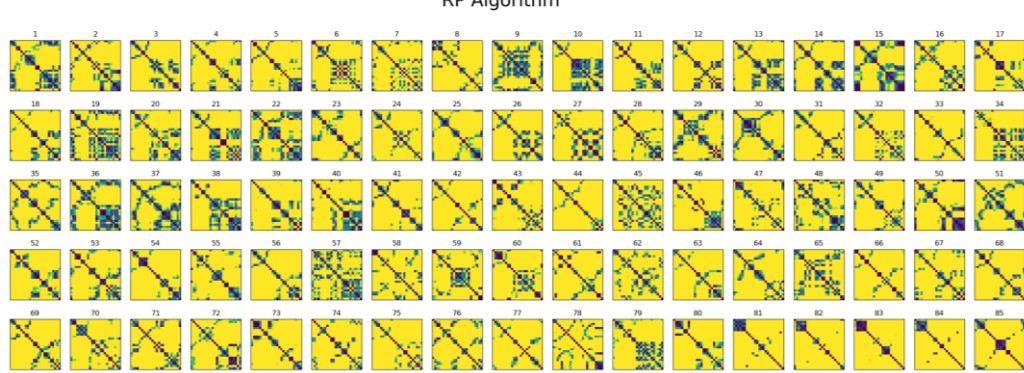

**Figure 10.** Time series image datasets recorded for 85 days (RP).

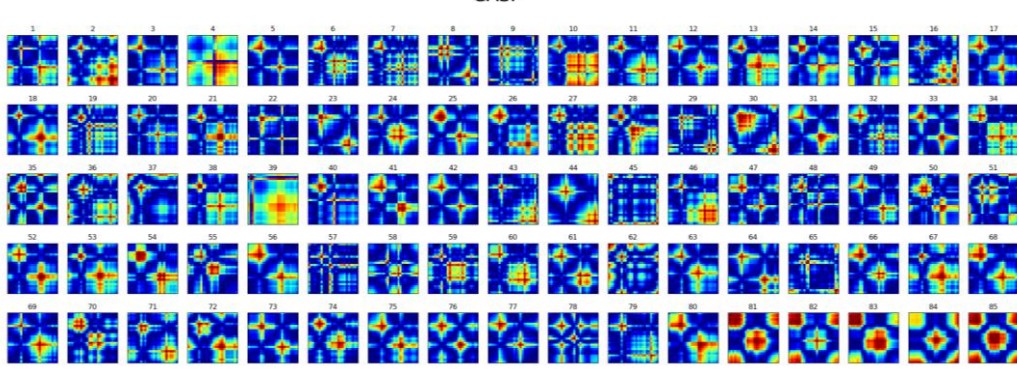

**Figure 11.** Time series image datasets recorded for 85 days (GASF).

GADF

**Figure 12.** Time series image datasets recorded for 85 days (GADF).

MTF

**Figure 13.** Proposed detecting anomaly method using time series images (MTF).

### *3.6. Using CNN-Based Model for Detecting Anomalies*

Although CNN is mainly used for object (defect, forgery, etc.) detection or image analysis, in this experiment [40,41], a convolutional neural network was used for binary classification of image datasets generated using four algor ithms. The neural network structure for binary classification is specified in Table 2. Since the shape of the image data used in this experiment was (24 and 24), the dimensions of the input data were set to (1, 24, 24). A neural network was constructed to determine whether the input image data were an anomaly through the neural network and to calculate a value close to 1 if normal and 0 if anomaly. After constructing two convolutional layers, a fully connected layer was placed for binary classification. The epoch was set to 50 to ensure sufficient learning. For the label for learning, the label constructed in the preprocessing stage of injecting anomaly was used, and the model was trained by dividing 70% of the training dataset and 30% of the test dataset to contain the same ratio of anomalies.

### 4. Result

In this experiment, the F1-score metric was used to indicate the accuracy of the model for detecting anomalies. To use a metric that shows higher accuracy, as the number of false positives and false negatives is reduced in the confusion matrix, the F1-score using the harmonic average of precision and recall was selected as the metric. Models for detecting anomalies for each cluster were separately configured and trained, and the F1-scores of the clusters were averaged and are shown in Table 3 to evaluate the performance of each method.

**Table 3.** Experiment result according to generating time series image and clustering algorithm.

|  | No Clustering | K-Means | K-Shapes | SOM |
|---|---|---|---|---|
| Cluster | 1 | 8 | 8 | 6 |
| **ARIMA** | 0.91 | 0.92 | 0.92 | 0.93 |
| **RP** | 0.94 | 0.94 | 0.93 | 0.94 |
| **GASF** | 0.93 | 0.93 | 0.93 | 0.95 |
| **GADF** | 0.94 | 0.94 | 0.94 | 0.95 |
| **MTF** | 0.96 | **0.97** | **0.97** | **0.97** |

As a result of the experiment, the clustering method using the SOM algorithm had the highest accuracy when pre-clustering for anomaly detection, and the accuracy was lower than the non-clustering method. In addition, after pre-clustering, the case of applying the CNN-based binary classification model by generating the time series data image obtained higher accuracy than the case of applying the ARIMA model using the time series data as it is. When the time series data image generation method was applied, the model to which the MTF algorithm was applied yielded the highest F1-score.

## 5. Conclusions

In this paper, we proposed a CNN-based anomaly detection model using time series images after pre-clustering to detect anomalies in time-recorded power data. In previous studies, anomalies were detected using a statistical model or an RNN-based unsupervised learning model to consider the statistical characteristics of time series data. The proposed method uses a pre-clustering technique to characterize time series data collected from various domains. In addition, to effectively detect anomalies, a CNN-based unsupervised learning model specialized for binary classification was used by converting time series data into time series image data. To use the proposed method, three methods of pre-clustering and four methods of converting time series data into images were used to compare performance.

For this experiment, data recorded and provided every hour from 1 June 2020 to 24 August 2020 were used, and data per hour were purified into data every three hours and normalized to perform clustering. The effectiveness of pre-clustering was proved by comparing the prediction accuracy according to the presence or absence of pre-clustering. In addition, other clustering techniques were compared to evaluate best pre-clustering techniques. K-means and k-shapes clustering techniques were conducted using the elbow method, and clustering was performed by setting appropriate neurons according to the number of observations to utilize SOM.

After pre-clustering, the power time series data written in this study was converted into an image form using recurrence plot, Gramian angular field, and Markov transition field algorithms, and anomaly data and normal data were discriminated using a CNN-based binary classification model. The performance of the anomaly detection model using all CNN-based time series image data were better than the performance of the anomaly detection method using the ARIMA model using the existing time series data. In particular, the performance of the binary classification model using the time series image dataset constructed using the MTF algorithm was the best.

For anomaly detection of time series data, pre-clustering the time series data and applying the anomaly detection model individually to each cluster showed higher performance than training the raw data on a single model. In addition, using a CNN-based unsupervised learning model by imaging time series data showed equivalent or better performance than using an ARIMA model using raw data. Therefore, to detect anomalies in time series data, it is also worth considering ways to augment or image the dimension of time series data. As a future study, we will build a dataset using other dimensional augmentation techniques as well as the imaging method used in this paper.

**Author Contributions:** Conceptualization, S.O. (Seungmin Oh); Project administration, T.-W.U.; Supervision, J.K. and Y.-A.J.; Writing—original draft, S.O. (Sangwon Oh); Writing—review & editing, J.K. and Y.-A.J. All authors have read and agreed to the published version of the manuscript.

**Funding:** This work was supported by Institute of Information & Communications Technology Planning & Evaluation (IITP) grant funded by the Korean government (MSIT) (2021-0-02068, Artificial Intelligence Innovation Hub), the Korea Electric Power Research Institute (KEPRI) grant funded by the Korea Electric Power Corporation (KEPCO) (No. R20IA02) and the MSIT (Ministry of Science and ICT), Korea, under the Innovative Human Resource Development for Local Intellectualization support program (IITP-2022-RS-2022-00156287) supervised by the IITP (Institute for Information & Communications Technology Planning & Evaluation).

**Conflicts of Interest:** The authors declare no conflict of interest.

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
