# Peer review of "Methods of Pre-Clustering and Generating Time Series Images for Detecting Anomalies in Electric Power Usage Data"

_electronics, doi:10.3390/electronics11203315_

Round 1
Reviewer 1 Report
1) There are minor but many grammatical mistakes which the authors must correct. Specially singular/ plural type mistakes. Line 2 in introduction is repetitive and can be deleted.
2) Authors must define the full forms or include a list of abbrevations. For example: SOM, RP, GASF, GADF, and MTF
3) The research design can be improved by including the specific uses of the clustered data. How does it help?
4) Testing the performance of clustered vs non-clustered data is not significant in terms of research objective, because it is obvious that a more organized data will give better accuracy over a non organized one, provided the data is meaningful and does not have too many outliers.
5) Authors must add a short para in introduction highlighting the contributions of the paper and how it adds to existing knowledge and how it is useful in practical grid applications.
6) Possibly some light must also be thrown on the possible usage of the clustered data, as showing a better accuracy is not a significant contribution.
7) It would be interesting to mention some devices that capture power usage data and describe what type of anomalies are normally present in power usage data.
8) Power data are preprocessed from raw data into various types of data. It would be nice to add some references which point at usage of image, financial and time series data in power sector.
9) Differences between the 3rd and 4th branch in Fig. 1 are not clear. (Discretization and modeling)
10) At the beginning of each section and sub-section, authors should add a line to mention what is explained within.
11) It is interesting to read that such a large dataset does not have any outliers. Table 1 cannot prove this alone, therefore authors are requested to include a boxplot to show the same, focusing on all the variables within the dataset.
12) What does 'Inertia of TS' refer to in Fig. 5?
13) Do the clusters in Fig. 6 correspond to any power data variable? It is not clear from the image, better labeling must be done
14) It is wierd that the pre-processing is presented after clustering. Infact the datasets must be checked first of all to check and report any missing data, anomalies, outliers and basic stats. In may experience, I have not seen any real dataset from any DSO which does not have such issues.
15) Even though I cannot believe the absence of anomalies in real data, but atleast the method for anomaly addition must be better justified and described. For example. random instances are fine, but the amplitude selection is not very convinciengly described. Is it white gaussian noise or just min-max based? What was the basis for such a choice?
Author Response
Comment #1
There are minor but many grammatical mistakes which the authors must correct. Specially singular/ plural type mistakes. Line 2 in introduction is repetitive and can be deleted.
Author Response
Thank you for your thoughtful consideration, down to the grammatical part. It was a good opportunity to look at the overall manuscript grammatical errors.
Author Action
Singular/plural grammatical errors have been checked throughout the manuscript. And, Lines 1 and 2 of the introduction have been modified to be seamlessly connected to the contents.
“ Most people living in modern civilization are provided with electricity produced by the state or private enterprise. From the perspective of the private company or government that supplies electricity, it is important to manage electric power received so that it could be used efficiently. ”
Comment #2
Authors must define the full forms or include a list of abbreviations. For example: SOM, RP, GASF, GADF, and MTF.
Author Response
Thanks for pointing out the abbreviations definitions in the section describing the relevant research and algorithms used.
Author Action
We have defined an explanation of the abbreviation where SOM, Rp, GASF, GADF, and MTF are first mentioned. We specified the definition in the subtitle by checking the first occurrence of the related term in the subtitle of Chapter 2 and at the beginning of each verse.
Comment #3
The research design can be improved by including the specific uses of the clustered data. How does it help?
Author Response
The electric power data used in this study is data used for anomaly detection and power supply planning. We pre-clustered the power data to show better anomaly detection performance. Since the data was recorded in the building, it has the characteristic that it can be grouped into the same cluster if it is a building that is engaged in the same business or built for the same purpose. Therefore, clustering this data is effective to analyze the recorded power data in the above situation. In future studies, it is expected that the effect of pre-clustering will be greatly enhanced if data labeled by business type is used for each building.
Comment #4
Testing the performance of clustered vs non-clustered data is not significant in terms of research objective, because it is obvious that a more organized data will give better accuracy over a non-organized one, provided the data is meaningful and does not have too many outliers.
Author Response
Thanks for the good point. In this study, rather than learning the original data as it is for anomaly detection, it has research significance in the approach of applying the anomaly detection model to each cluster individually through pre-clustering. In an anomaly detection and power pattern analysis study using existing power data, we learned from randomly collected power data without prior clustering. The data used in this study is the data actually detected in 60 buildings, and in practice, data from such an environment is frequently used. Therefore, this paper says that the approach to detecting anomalies by introducing pre-clustering in real situations is effective.
Comment #5
Authors must add a short para in introduction highlighting the contributions of the paper and how it adds to existing knowledge and how it is useful in practical grid applications.
Author Action
Thanks for the feedback on the field of utilization. Added relevant information at the end of the introduction.
“If the approach proposed in this paper is used, we can show high performance in electric power analysis tasks such as anomaly detection of power data in various electric power detection systems. For example, in a smart grid, a pre-clustering-based power anomaly detection method can be used to have a generation-optimized electric power supply pat-tern. Moreover, the method of this study can be adapted to label the generation in which anomalies are recorded.”
Comment #6
Possibly some light must also be thrown on the possible usage of the clustered data, as showing a better accuracy is not a significant contribution.
Author Response
Thank you for your feedback on the research value of this thesis. As mentioned in the previous comment, the power data used in this study is actually electric power data collected by KEPCO(Korea Electric Power Corporation). Data collected in real situations may have different power patterns depending on the characteristics of the generation being detected. We proposed a method of pre-segmenting data to analyze power patterns by performing pre-clustering using only the detected data. Even if it is not for anomaly detection, we believe that pre-clustered data can be applied in various ways such as prediction, latent data generation, and data augmentation through pre-clustered power data.
Comment #7
It would be interesting to mention some devices that capture power usage data and describe what type of anomalies are normally present in power usage data.
Author Action
Thanks for the feedback on the power usage data. Added examples of power usage data in Chapter 2 2.1 of the main text. And I explained how outliers are recorded in a typical detector.
“ Figure 1 is an example of normalized power data actually detected in a building. We can clearly see the trend in electric power usage from day to day, and if we take a long time unit as a basis, we can see the trend in that electric power usage. This data is record-ed by a general building detector, and if the detector malfunctions due to various environmental factors, the maximum or minimum value of the measured power value may be recorded abnormally.”
Comment #8
Power data are preprocessed from raw data into various types of data. It would be nice to add some references which point at usage of image, financial and time series data in power sector.
Author Action
Thank you for your feedback on the recording of various forms of power data. In the second paragraph of this 2.1, manuscript includes contents that electric power data is pre-processed from raw data to various types of data. We have added representative example data for each type.
“For example, we can use a picture of a detector or a faulty power plant equipment as im-age data.”
“Financial data includes data that adjusts unit price to plan power supply according to the recorded amount of electric power.”
Comment #9
Differences between the 3rd and 4th branch in Fig. 1 are not clear. (Discretization and modeling)
Author Response
Thanks for the feedback on Figure 1 (now figure 2). The third branch in this figure means that it takes the place of the task of clustering by applying a model that extracts features in tasks that generally require clustering. As for the clustering of the time series dataset, which is mainly explained in this figure, the fourth branch is clearer. The fourth means to use a general-purpose clustering technique by using the coefficients and residuals that are calculated after applying a general-purpose model to the dataset.
Comment #10
At the beginning of each section and sub-section, authors should add a line to mention what is explained within.
Author Response
Thank you for taking a look at the structure of the thesis. After reviewing the manuscript, we decided that it would be appropriate to add a text to the section at the beginning of sections 2 and 3. So, as in this comments, we added that to section 2, 3.
Comment #11
It is interesting to read that such a large dataset does not have any outliers. Table 1 cannot prove this alone, therefore authors are requested to include a boxplot to show the same, focusing on all the variables within the dataset.
Author Response
Thanks for pointing out this dataset. The data used in this study is a dataset provided by KEPCO (Korea Electric Power Corporation). Before open access, a data set with no outliers among the building power usage data sets is selected and shared data, so there are no critical outliers.
Author Action
A boxplot was drawn from one of our datasets and added to section 3.1. An additional description of the dataset is given in Section 3.1.
“Figure 4 shows a boxplot for each column in the dataset. Outliers were recorded in the wind speed column and humidity column in all buildings, and few outliers were found in power usage or temperature.”
Comment #12
What does 'Inertia of TS' refer to in Fig. 5?
Author Response
The metric representing the sum between clusters is called inertia. To be more specific, since it is the inertia for the Total Sum of squared errors between clusters, we described it as the Inertia of TS.
Comment #13
Do the clusters in Fig. 6 correspond to any power data variable? It is not clear from the image, better labeling must be done.
Author Response
Thanks for pointing out Figure 6. What is depicted in the figure is the result of the K-Shape Clustering algorithm, and electric power usage data included in each cluster and a representative trend line are depicted.
Comment #14
It is weird that the pre-processing is presented after clustering. In fact, the datasets must be checked first of all to check and report any missing data, anomalies, outliers and basic stats. In may experience, I have not seen any real dataset from any DSO which does not have such issues.
Author Response
Thank you for your feedback on the overall flow of the experiment. Data preprocessing referred to in section 3.3 refers to the procedure performed to input clustered time series data into an anomaly detection model. In general, it refers to preprocessing in the form of segmenting train and test datasets or transforming dimensions for imaging. The preprocessing required to implement clustering has been completed before that.
Comment #15
Even though I cannot believe the absence of anomalies in real data, but at least the method for anomaly addition must be better justified and described. For example. random instances are fine, but the amplitude selection is not very convincingly described. Is it white gaussian noise or just min-max based? What was the basis for such a choice?
Author Response
In general workplace power detectors, outliers are recorded due to power outages, equipment errors, short circuits, etc. Usually in such cases the minimum or maximum value of the power is recorded. Therefore, we adopted a min-max-based outlier injection method. As mentioned in the previous comment, we were provided with a curated dataset from our power supplier, so we were able to use a dataset that had fewer outliers and was of good quality.

Reviewer 2 Report
This work is devoted to the actual topic of detecting anomalies in electricity data. Currently, accurate metering and control of the quality of electricity are of great importance in the relationship between the supplier and the consumer. Every year, artificial intelligence penetrates more and more into the energy sector and is widely used to solve various problems there. The paper proposes a CNN-based anomaly detection model using image time series after pre-clustering to detect anomalies in time-recorded power data. The developed model was compared with the already used model and showed the best result. There are some remarks about the literature used in the list of sources, 50% of which is older than 5 years, and 35% older than 10 years. For such a new hot topic, there are more recent studies to analyze and compare. I would like to see more new works in the list, please. Despite this remark, in general, the article has a high scientific and practical significance, is relevant and deserves to be published in this journal after minor revision.
Author Response
Thank you for your thoughtful comments about this manuscript. This study has significance in applying various imaging algorithms designed 5-10 years ago to analysis and pre-clustering of time series data, rather than a relatively recently devised algorithm. Therefore, it is not easy to find cases where research papers on recent trends are referenced. Please understand this.
Round 2
Reviewer 1 Report
None